# Cancer Healthcare Workers’ Perceptions toward Psychedelic-Assisted Therapy: A Preliminary Investigation

**DOI:** 10.3390/ijerph18158160

**Published:** 2021-08-02

**Authors:** Lisa M. Reynolds, Amelia Akroyd, Frederick Sundram, Aideen Stack, Suresh Muthukumaraswamy, William J. Evans

**Affiliations:** 1Department of Psychological Medicine, The University of Auckland, 22-30 Park Avenue, Grafton, Auckland 1023, New Zealand; m.akroyd@auckland.ac.nz (A.A.); f.sundram@auckland.ac.nz (F.S.); asta123@aucklanduni.ac.nz (A.S.); 2School of Pharmacy, The University of Auckland, 85 Park Road, Grafton, Auckland 1023, New Zealand; sd.muthu@auckland.ac.nz; 3Mana Health, 7 Ruskin Street, Parnell, Auckland 1052, New Zealand; info@manahealth.co.nz

**Keywords:** psychedelic, cancer, perceptions, qualitative, healthcare

## Abstract

Recent clinical trials suggest that psychedelic-assisted therapy is a promising intervention for reducing anxiety and depression and ameliorating existential despair in advanced cancer patients. However, little is known about perceptions toward this treatment from the key gatekeepers to this population. The current study aimed to understand the perceptions of cancer healthcare professionals about the potential use of psychedelic-assisted therapy in advanced cancer patients. Twelve cancer healthcare professionals including doctors, nurses, psychologists and social workers took part in a semi-structured interview which explored their awareness and perceptions toward psychedelic-assisted therapy with advanced cancer patients. Data were analysed using thematic analysis. Four inter-connected themes were identified. Two themes relate to the role and responsibility of being a cancer healthcare worker: (1) ‘beneficence: a need to alleviate the suffering of cancer patients’ and (2) ‘non-maleficence: keeping vulnerable cancer patients safe’, and two themes relate specifically to the potential for psychedelic-assisted therapy as (3) ‘a transformative approach with the potential for real benefit’ but that (4) ‘new frontiers can be risky endeavours’. The findings from this study suggest intrigue and openness in cancer healthcare professionals to the idea of utilising psychedelic-assisted therapy with advanced cancer patients. Openness to the concept appeared to be driven by a lack of current effective treatment options and a desire to alleviate suffering. However, acceptance was tempered by concerns around safety and the importance of conducting rigorous, well-designed trials. The results from this study provide a useful basis for engaging with healthcare professionals about future research, trial design and potential clinical applications.

## 1. Introduction

Advanced-stage cancer is commonly associated with psychiatric disorders including high rates of treatment-resistant depression and anxiety [1,2], and psychological distress more broadly including existential despair, loss of meaning and desire for a hastened death [3,4]. Psychological problems such as these are not only associated with poor quality of life [2] but also with poorer physical health outcomes such as increased pain [5], decreased treatment adherence [6], prolonged hospitalisation [7], and increased mortality [8]. Unfortunately, the efficacy of standard treatment for these kinds of psychological problems is mixed and limited in advanced cancer populations. Recent interest in psychedelic-assisted therapy suggests it may be a promising new approach which may offer benefit [9]. However, little is known about perceptions toward this treatment from the key gatekeepers to this population, i.e., cancer healthcare professionals.

The standard treatment offerings for depression, anxiety, and issues such as existential despair are limited. Several meta-analyses of pharmacotherapeutic interventions have failed to demonstrate clear benefit in cancer patients [10,11,12] and typical medications such as antidepressants and anxiolytics are associated with delayed onset of clinical improvement, high relapse rates, and adverse effects which can impact safety and compromise treatment adherence [13]. Likewise, psychological interventions in cancer contexts can be problematic with low participation, high attrition [14] and debatable effectiveness [15]. A recent review found that combining psychopharmacology with psychotherapeutic approaches can lead to outcomes superior to either treatment alone in depression [16]. The benefit of combining such therapeutic modalities is increasingly being recognised within the disciplines of palliative care and psycho-oncology [10].

One combined approach that appears to offer particular promise lies in administering psychedelic compounds alongside psychotherapy [17]. The theoretical basis of this approach lies in the fact that classical serotonergic psychedelics, including psilocybin and lysergic acid diethylamide (LSD), appear to stimulate 5-HT2A receptors in the brain [18] and have been shown to increase synaptic function [19] and proteins that are associated with neural plasticity (Brain Derived Neurotrophic Factor, BDNF; [20]). Evidence suggests that 5-HT2A receptors are centrally involved in affect regulation [18]. Brain plasticity of this kind has been reported as a key mechanistic pathway in antidepressants [21] and has long been argued as a potential catalyst for psychotherapeutic processes [22]. Recent work has also found that psychedelics can enhance suggestibility [23] and meaning in life [24]. Thus, it seems feasible that the administration of psychedelics might create a neural landscape where a person is more receptive and better able to absorb the new ideas being presented in psychotherapy.

Although several studies combining psychedelics with psychotherapy were conducted with cancer patients in the 1960s and 70s [25,26,27], it has only been in the last decade that an interest in this area has re-emerged. While few in number and limited in design, the quantitative studies with cancer patients who have depression and/or anxiety in this area have all found sustained reductions in depression, anxiety, and existential despair and increases in quality of life [28,29,30]. A recent review of relevant clinical trials reported that a single psychedelic experience can produce rapid, robust, and sustained improvements in cancer-related psychological distress [9]. Similarly, a qualitative investigation with cancer patients who had participated in psychedelic–assisted therapy reported that participants described enduring positive changes including an enhanced sense of meaning and less fear of death [31]. Although research in this area is still at an early stage, such fast-acting and enduring responses are of particular interest in the palliative context where treatments need to work both effectively and quickly.

Despite research investigating psychedelic agents reporting only mild or transient side effects (e.g., increased blood pressure and heart rate, anxiety, headaches and nausea [18]) and conditions for safe administration having been established [32,33,34], there are several reasons to suspect that cancer healthcare professionals may have concerns about this approach. First, there have been isolated reports in the scientific literature about adverse effects including delirium or temporary worsening of psychological distress [35]. There have also been widespread accounts in the media about the dangers of psychedelic-induced violence and psychosis [36]. Furthermore, that these compounds are almost universally classified as Schedule 1 drugs with “no currently accepted medical use” and “a high potential for abuse” [37] may also give rise for concern. Given that cancer healthcare professionals are important stakeholders (and gatekeepers) in the healthcare and wellbeing of their patients, understanding their perceptions and potential concerns is important. Apart from limited studies investigating attitudes of psychiatrists and palliative care experts toward psychedelics [33,36], no studies have specifically investigated the perceptions of cancer healthcare professionals. The objectives of the current work were to investigate the awareness, knowledge, attitudes, and perceptions of cancer healthcare workers in relation to the potential use of psychedelic-assisted therapy with advanced cancer patients who have depression and/or anxiety.

## 2. Materials and Methods

### 2.1. Researcher Declaration

The researchers involved in the study are health psychology practitioners (AA, LR, AS), medical clinicians (FS, WE), and academic researchers (SM) who have a particular interest in the intersection between psychology and medicine. One of the researchers (LR) had a professional relationship with some participants in this study, however, this researcher was not involved in the data collection for these participants and the data were de-identified before analysis to help limit potential bias.

### 2.2. Participants

Participants were eligible to take part if they were proficient in English and provided healthcare treatment or support to patients with advanced/metastatic cancer. Previous research suggested differences in attitudes towards psychedelics between younger vs. older, men vs. women, and trainees vs. more experienced clinicians [38], therefore recruitment was targeted to ensure a spread of participants across these demographics and professional variables.

### 2.3. Study Procedure

In order to minimise recruitment bias, the study was advertised in neutral terms as an interview “exploring a new way of supporting patients with advanced cancer” rather than an investigation of perceptions about psychedelic medicine. A variety of recruitment methods were utilised including email invitations to relevant cancer services and flyers presented at clinical meetings. Interested participants were asked to contact the researchers to be screened for eligibility. Recruitment ended following data saturation and when there was a sufficient range of age, gender, ethnicity, professional roles, and years of experience. The study was approved by the Health and Disability Ethics Committee (19/STH/122) and the Auckland District Health Board (A + 8576) and all participants completed written consent before commencing the study. Participants were offered a gift worth approximately NZD 25 for their involvement in the study.

Interviews took place between October 2019 and January 2020 and were conducted in a location of the participant’s choice. Three participants were interviewed at their workplace, six at the University, two over the phone, and one at home. Two researchers (AA, LR) completed the interviews. The interviews followed a semi-structured format where questions began broadly by asking participants to describe their professional role and their observations about how their patients coped with advanced cancer before moving to specific questions about psychedelics. Discussion centred around various terms written on cards including ‘psychedelics’, ‘hallucinogens’, ‘LSD’, ‘psilocybin’, ‘magic mushrooms’, ‘ayahuasca’, ‘mescaline’, ‘ibogaine’, ‘MDMA’, ‘micro dose’ and ‘high dose’. Next, a brief description of recent studies was presented (see Figure 1) and participants were encouraged to express their first impressions, concerns and perceptions of benefits. Participants were reminded about confidentiality and encouraged to speak freely. The interviews ranged from 30 to 59 min in length and were recorded using a Sony ICDPX470 Digital Voice Recorder. The interviewer also wrote notes based on observations from the interview.

### 2.4. Data Analysis

A qualitative approach was undertaken given the exploratory aims of this work. A critical realist position was taken in analyses where literal meaning was balanced with interpretation of meaning through a biopsychosocial lens [39]. Data were analysed using inductive reflective thematic analysis which encourages rigorous analyses of the data while also allowing the researchers to gain rich and meaningful insights [40]. The interviews were transcribed verbatim by an independent external source and data checked for accuracy before thematic analysis by the research team. Analysis followed six steps. 1. *Data familiarization*: listening to recordings, reading transcribed interviews, and making general notes. 2. *Coding*: labelling quotations in the interview data to produce codes. 3. *Generating initial themes*: identifying potential themes in the data. 4. *Reviewing themes*: using NVivo12 to organise and allocate quotations. 5. *Defining and naming themes*: the meaning and boundaries of themes were clarified, and themes were labelled. 6. *Writing report*.

## 3. Results

Twelve healthcare professionals working in either cancer care or palliative care settings across the Auckland/Northland regions in New Zealand took part in the study (see Table 1). The sample had a mean age of 40.5 years (SD = 11.9) and the majority were female (75%), New Zealand European (58%), and doctors (42%). The mean years of experience was 12.08 years (SD = 10.19).

Before discussing the key themes related to the particular focus of this work, it is useful to note some general observations that emerged from our interviews. First, in beginning our conversations by asking about professional responsibilities and experiences with patients, participants were open and talked freely about their experiences. However, it is noteworthy (although perhaps not unexpected) that when questioning turned to psychedelics, participants often expressed initial surprise and exhibited subtle body language that suggested some hesitation about the subject matter. This hesitation appeared more obvious when participants were interviewed in their workplace. Initial comments such as “*probably the first thing* [that comes to mind] *is recreational drugs*” [Doctor, 1], and “*I guess I know very little about their use outside of recreational drug use*” [Social Worker, 3] suggest hesitation or discomfort may have been related to their association with illicit use. However, hesitation was short-lived in all interviews, and participants moved on to talk freely about their knowledge and perceptions of psychedelics from a professional and (sometimes) personal perspective.

Another general observation relates to a notable range of knowledge and awareness across participants. Some participants admitted to knowing very little about the topic—“*with the psychedelics like, I think cocaine is in there eh? … I don’t know, I’m not really knowledgeable really*” [Psychologist, 8]. In contrast, others claimed to have good knowledge:
“*Yeah, I know a lot … in the last palliative care conference there were excellent presentations … they work with LSD in palliative care, it’s fascinating, and has been shoved under the carpet because of the Vietnam war and I don’t really know the whole history … could talk for hours about that*”.[Doctor, 12]

When participants had limited knowledge in the area they seemed more likely to rely on a heuristic informed by information gained through various sources—“*sort of glimpse of that rather than anything else and the rest of it is not actually from my experience, its more from TV and on the news actually*” [Doctor, 1]. Examples of sources that informed their view were the media (e.g., Netflix documentaries), literature (e.g., Aldous Huxley), and through personal experience, e.g., “*you know personal experience of being at university, you know stories that you hear*” [Psychologist, 7]. Typical associations were with hippies and the 1960s and 1970s, e.g., “*I guess I think back to the 70s, psychedelics was a term that was used quite a lot in the sort of hippy, free living days*” [Psychologist, 4]. Participants that had greater knowledge about psychedelics through scientific or professional forums seemed less likely to draw on anecdotes, media reports or personal experience. Thus, it is useful to consider that perceptions are almost certainly likely to change as knowledge increases across healthcare professionals and more generally.

### 3.1. Key Themes

The key focus of this work was to explore the perceptions of cancer healthcare workers about psychedelic-assisted therapy. Four inter-connected themes related to this focus were identified and the relationship between these themes is conceptualised in Figure 2. Two themes related to the role and responsibility of being a cancer healthcare worker: ‘beneficence: a need to alleviate the suffering of cancer patients’ and ‘non-maleficence: keeping vulnerable cancer patients safe’. A further two themes related specifically to psychedelic-assisted therapy as ‘a transformative approach with the potential for real benefit’ but that ‘new frontiers can be risky endeavours’. The themes are described below with example quotes from participants.

#### 3.1.1. Theme I: Beneficence—A Need to Alleviate Suffering

All participants described the particular suffering that advanced cancer patients face including dealing with complex and physically demanding treatment regimens, coping with challenging emotions such as fear and anxiety, negative cognitions like worries about dying, and unhelpful avoidance behaviours or denial.

“*There are people that I see that aren’t coping well*” and “*I think there is a lot of avoidance and they try to avoid the emotions that they are feeling and I guess live in denial. So I think it definitely probably increases stress, more avoidance, more anxiety among those patients which is really difficult, because I think when it is advanced cancer you want them to be able to live the rest of their days the best that they can … at the end of the day advanced cancer patients are in a really difficult situation*”.[Psychologist, 6]

Participants described that their role as healthcare workers is to try to alleviate this suffering. One psychologist simply stated: “*I want to take away the pain*” and “*you know when you have patients that are really feeling the pain of chemo treatment and you think to yourself, God, I’ve got the opportunity here that if they could take this and it will alleviate perhaps the mental stress*” [Psychologist, #8].

However, participants also acknowledged that there were times when the suffering of patients could not be alleviated “*if you are asking could there be more, then, yes absolutely, and especially I have patients who have struggled a lot*” [Doctor, #1], “*I don’t think we ever really, truly, alleviate all their distress as much as we would like to*” [Psychologist, 6], and “*a patient who’s very practical and she goes, you know, I’ve done my advanced care plan, I’ve got everything in order, but I’m still afraid of dying.*” [Psychologist, #8].

The alleviation of suffering could also be frustrated in the context of current treatment approaches that are not always helpful or fall short in addressing patient need:
“*I think probably, there is a big unmet need … I think that often we, because it is easier, we focus on our job of discussing treatment, and obviously we are aware of how distressed people are, but we aren’t always able to address it appropriately*”.[Doctor, #2]

In such situations, feelings of helplessness in the healthcare workers often ensued:
“*It can be really difficult. I think, you know, as a health professional you want to be compassionate, you want to alleviate someone’s suffering, and so you feel bad that you can’t get them out of that situation … It can be really difficult knowing that even though someone has got such a short time to live they’re still suffering. It is really difficult*”.[Doctor, #2]

It followed that participants were generally open to an approach that might alleviate suffering in their patients—“*you know, I would do most things, within reason, to help them improve their quality of life. So, I wouldn’t say I’m against it*” [Psychologist, #6]. Likewise, they also reported that patients themselves would probably be open to such an approach:
“*There are people that I see that aren’t coping well and I think that something like that, if it was going to increase their ability to be open and, I think, alleviate some of the anxiety and depression so that they can engage in a life that they want to for the rest of their lifetime that they have, then I think that they would be willing to consider it*”.[Psychologist, #6]
“*When you have advanced cancer you are willing to do anything to improve your life, and so I think in one way you know it’s not curing the cancer, but it does give some people hope that they will be able to live a better life throughout those months*”.[Social Worker, #5]

In summary, this theme captures a professional responsibility and a desire in healthcare workers to alleviate suffering in a population where suffering can feature so intensely and is not always effectively ameliorated.

#### 3.1.2. Theme II: A Transformative Approach with the Potential for Real Benefit

All of our participants noted that the concept of incorporating psychedelics in healthcare stretched the current medical paradigm through a transformative approach that had potential to offer real benefit.

“*Something which society, or the powers that be, have deemed as illicit and not good for humans could potentially be slightly altered and given to people in an appropriate dose to make their health or whatever better … that’s part of science, you kind of have to push the boat out a bit … that’s what research is for*”.[Doctor, #9]

However, one participant questioned whether society was ready for such an approach*—*“*can New Zealand society handle the concepts that this type of research is going on? Is New Zealand society mature enough to?*” [Psychologist, #6].

Interestingly, there was a difference in perspectives between people who had prior knowledge of psychedelic use as a therapy and those that did not. Participants who had limited prior knowledge generally expressed initial hesitation. Comments in the early stages of interviews were sometimes guarded—“*I’m not sure how that would work with people who are in the advanced stage of life*” [Psychologist, #7], “*initially, oh my gosh, like that seems crazy*” [Psychologist, #4] and “*I guess that one of my biggest concerns is that there doesn’t seem to be a lot of evidence*” [Psychologist, #6]. However, after consideration and hearing about results from recent studies a cautious curiosity emerged—“*I’m willing to give things a go, but it needs to be common sense … plus, it needs to be evidence-based*” [Social Worker, #3] and some people switched to a more open and supportive view—“*that’s the whole point of research, that we test out these interesting and potentially weird and wacky ideas because you know, we might find something that works and is helpful*” [Psychologist, #4].

In contrast, where participants had prior knowledge about psychedelics, they seemed more immediately open to the idea. In our study, it was doctors who had greater knowledge about the use of psychedelics as a medicine. One doctor commented that “*I’ve always been convinced that they have a major place and that good research is necessary*” [Doctor, #11] and another said:
“*I think that there is a whole branch of psycho-pharmacology that’s been latent for 50 years and it’s just scientifically interesting, where could it lead, what could it mean? … that interests me as a scientist. It might not work, it might turn out to be too toxic, but it’s there waiting to be investigated and I think the investigations have been shut down for the wrong reasons*”.[Doctor, #12]

Building on the need to alleviate suffering, participants could see a role in the context of advanced cancer:
“*The idea of patients feeling like they need to process … their existential concerns, in terms of what has my life meant … what happens from here … what do I believe in … do I die and that’s it, the end of me. So … I guess in that sense, an experience like this would open their mindset enough to maybe explore some of those existential ideas that typically comfort patients when they have advanced cancer*”.[Psychologist, #4]

There were also comments demonstrating the idea that a medication that impacts neural activity seemed like an intuitive mechanism of change “*It just makes sense to me. If you’re clinically depressed, then a drug that is going to … allow your synapses to awaken and … change your mood, then I’m all for that.*” [Nurse, #10].

Although the idea of using psychedelics in a medical context was perceived as innovative in the context of the West, one participant highlighted the fact that they have a long history in many indigenous cultures:
“*Ibogaine, ayahuasca, mescaline have got a fairly long traditional history … so, there’s a large volume of traditional knowledge … I think that the ones that have got the history are also in the bed with the large body of traditional and indigenous knowledge, and so they intersect with quite different worlds or world views of understanding*”.[Doctor, #12]

This same participant cautioned against exploitation of such indigenous knowledge and the spiritual aspects of such an approach:
“*I worry about such research that it trivialises the spiritual dimensions of stuff like this … one thing that pops out at the start is worries about cultural appropriation … because we don’t really understand the compound … it’s just the leaping off point for a whole cultural understanding, which you can’t really separate from the compounds*”.[Doctor, #12]

Thus, although much of the commentary from participants suggested that psychedelics might be a transformative approach in the context of Western medicine, words of caution and a recognition of indigenous knowledge were also noted.

#### 3.1.3. Theme III: New Frontiers Can Be Risky

Alongside perceptions about potential benefits were also perceptions of risks. In the absence of evidence (or knowledge about evidence), participants drew on various sources to form an amalgam of perceived risks. These included problems with being associated with “illegal” and “recreational” drug use:
“*I think with substances that are illegal, they are not evidence-based, and I think that if something’s not evidence-based then it’s not good to consume it …*”[Psychologist, #6]
“*It’s portrayed as something that is used for fun, rather than for medicinal use and I guess, if I was to think about it in a medical context, rather than in a fun context, that seems to jar—like things that you do for fun aren’t often things that are prescribed*”.[Psychologist, #4]

Associations with illegal use generated concerns about the manufacturing process*—*“*production of it will happen through some sort of unregulated means, which I think means that there might be risks associated with taking it, not purely because of the substance but because of how the manufacturing process and that sort of thing.*” [Doctor, #9].

Perceptions of risks included “*drug induced psychosis*” and “*it just makes me feel like it’s something you wouldn’t necessarily be able to control*” [Doctor, #2], being “*bad for you, leading to poor health outcomes, risky behaviours, I mean possibly death*” [Psychologist, #4] and that someone might take a psychedelic “*into their home … doing it in not such a safe way, is probably the biggest thing*” [Psychologist, #6]. There were also concerns about altered states of consciousness—“*what happens in that altered state … I suppose the concern that … [it] would cause a psychotic episode for some vulnerable people.*” [Social worker, #5] and the negative impact of ‘bad trips’ or being put in dangerous situations “*bad trips or psychosis, not being safe, physically safe, and putting yourself in danger—yeah.*” [Social worker, #3].

Of particular relevance to advanced cancer patients, who are typically on many other medications, was a concern about the possibility of drug interactions “*I would worry about people being on active treatment and drug interactions.*” [Doctor, #2] and “*it would depend in what context it was being used … I guess if it were being used in like a cancer environment then I would be a bit like, gosh, is this appropriate?*” [Psychologist, #4].

A closing note of caution about the potential potency of psychedelic treatment was provided by one doctor who stated: “*The feeling that I have is similar to the feelings that I have about high voltage electricity, it’s really important … and potentially very powerful for all the things that we do … and you have to be very careful around it*”.[Doctor, #12]

#### 3.1.4. Theme IV: Non-Maleficence—Keeping Vulnerable Patients Safe

In line with the professional role to alleviate suffering, and concerns about risk, were numerous comments about the responsibility to not cause harm. This responsibility is of particular relevance in the context of a vulnerable advanced cancer patient being provided a treatment approach which is perceived as unproven and a compound that has “*negative associations*” [Doctor, #1]. Although participants were generally open to the idea of psychedelic medicine, concerns were raised about the risks of using such substances with such a population:
“*We’re talking about … a vulnerable population or people that are at risk. So, I’m yeah—that is my concern—I mean perhaps there has to be more research with people who don’t have cancer before advancing to people that do have cancer*”.[Doctor, #12]

However, participants were also quick to offer suggestions about how safety concerns might be managed including building a scientific evidence base, researching psychedelic compounds in the same way that other drugs are researched, and considerations of dosage.

Participants recognised the importance of evidence-based practice and the necessity to build that evidence through well designed and rigorously conducted research trials. Psychologist #6 stated that reassurance about usage would come from scientific evidence “*if something was offered that had proven effects that was done in a controlled environment by people who have researched it, then I don’t think I would have an issue with it*” and another psychologist #4 noted that medically controlled environments imbue safety—“*it’s quite a controlled environment so minimises the risks of those things happening.*” Other participants suggested that, as long as there was supporting evidence, psychedelic medicines could become just another pharmaceutical agent:
“*They are just another drug, so as long as there is rigorous testing like there is with all of the drugs that people are being given now, if there is evidence to support, I personally wouldn’t have a problem with it*”.[Social worker, #3]
“*You’ve got to have a good template for understanding the toxicity, that’s the way we develop cancer drugs—with the first step is understanding, it’s understanding the toxicity of the medicine. So, you know that’s why we’ve got Phase One studies to learn what the limits are on dosing, what the dosing toxicities are, and the sort of things that can go wrong. This is all very well parameterized, protocolised, in how you develop medicines*”.[Doctor, #12]

Responses related to safety were also made in connection to the idea of dosage. Participants noted that a smaller dose (i.e., a micro dose) would be associated with less risk of side effects and there was comfort in this lower risk profile. “*The fact that it, the idea of micro-dosing, of it being a small dose, it feels/makes it seem more acceptable*” [Psychologist, #4] and “*any concentration of something, as it goes up, the propensity for it to have adverse effects increases, so yeah, so referring someone to a micro-dose treatment or a trial, I would feel more comfortable with that.*” [Doctor, #9]. However, a comment was also made that although a micro-dose would be less risky, it might also be less effective: “*The idea of micro-dosing, so very small doses which I take to mean doesn’t necessarily induce all of those psychedelic effects that a full dose would…but then I guess the question is what’s the point in this micro-dose?*”[Psychologist, #4]

Thus, although participants expressed concerns about the risks of psychedelic-assisted therapy, this theme captures perceptions of well-established practices to keep patients safe through a process of incremental research and phased clinical trials.

## 4. Discussion

Despite the widely acknowledged ‘renaissance’ in interest and research regarding the potential for psychedelics to address both psychiatric dysfunction [17,41,42] and cancer-related distress [9], there has been scant attention to what cancer healthcare workers (or, for that matter, patients) might make of this treatment. Given the stigma and misconceptions toward psychedelics in general [43], understanding the perceptions of healthcare workers toward psychedelic-assisted therapy in cancer contexts seemed like an important foundational step before extending work in the area. The current study identified perceptions that fell under two overarching themes (1) the responsibility for beneficence and non-maleficence in cancer healthcare workers and that (2) psychedelic-assisted therapy appears to be a transformative approach with potential for real benefit but also high risk. Below, we consider these themes in the context of relevant literature, discuss study limitations, and propose an agenda for future research in the area.

### 4.1. The Responsibility of Healthcare Workers: Beneficence and Non-Maleficence

Advanced cancer patients are a population with complex vulnerabilities [44] and it was clear that the participants in our study were well aware of their duty of care. Living with advanced cancer comes with physical and psychological challenges which are often compounded by social difficulties [45]. Consistent with the extant literature [3,4], our findings indicate a need for new ways of supporting this population. Our participants clearly articulated a sense of professional responsibility to alleviate this suffering balanced with the responsibility not to cause harm. Such themes of benefit and harm fit within the broader healthcare principles of beneficence and non-maleficence [46] and these principles appeared to provide an important filter through which healthcare workers appraised the concepts we discussed.

Importantly, there is a sensitivity to balancing benefit and harm in providing healthcare and where that balance may land with one group may be different to another. Although we did not specifically ask participants in our study, it seems likely that the parameters of cost versus benefit decision-making differ at the end of life compared to other contexts; what is appropriate in one context (for instance, early stage cancer) may be different to another (late stage cancer). In advanced cancer, patients are well versed on balancing therapeutic value with potential side effects (see below) and their context, in which a shortened life looms closely, matters. Thus, it is important we consider this context when making decisions about what might be appropriate in healthcare.

### 4.2. Perceptions about Psychedelic-Assisted Therapy: A Transformative Approach That Is High Risk/High Reward

The results of the current study suggest that cancer healthcare workers are optimistic about the possible benefits of psychedelic approaches but are also mindful of adverse effects. The judicious balance between risk and reward in therapeutic treatment is not new. In the cancer context there are numerous examples of treatments that, alongside healing, can cause unwanted symptoms such as vomiting, nausea, peripheral neuropathy and neurotoxicity (see chemotherapy side effects; [47]), psychiatric medications that can trigger mania, impact libido, and promote tumour growth [48], and even ‘softer’ therapies like psychotherapy are not the benign treatments that some people may think they are (see suicidal ideation and worsening distress; [49]).

One of our participants noted that we already have a pathway for navigating this risk and reward dichotomy, i.e., well designed and rigorously conducted phased research trials. That decisions about treatment implementation be informed by high-quality scientific evidence is well established in clinical forums [50]. Thus, the voice of our participants provide a further mandate for conducting well-designed clinical research trials in the area of psychedelic-assisted therapy in advanced cancer contexts. However, we would also argue that alongside evaluating treatment outcomes with the traditional gold standard approach to research (i.e., randomised controlled trials), complementary approaches such as pragmatic trials [51] which assess the real-life clinical implications of treatment should also be considered. In a treatment such as psychedelic-assisted therapy such flexible approaches to research may provide important insight.

Our findings also highlighted a range of knowledge about psychedelics across cancer healthcare workers and that this knowledge base influences perceptions. Prior knowledge of the psychedelic literature was associated with greater openness to the potential for psychedelics to play a role in Western medicine. Whilst the fact that knowledge might impact perceptions is unsurprising and fits with other research [52], it is nonetheless noteworthy in the current context where the topic of psychedelics is frequently being presented in the news, social media, movies and the like. Even within the brief context of a 30–60 min interview, perspectives became noticeably more open as the interview progressed, seemingly through a process of talking about the medical use of psychedelics. Thus, as the therapeutic potential of psychedelics is becoming more widely discussed and the landscape of awareness is rapidly changing, knowledge and perceptions will inevitably change.

### 4.3. Study Limitations and Research Agenda

The current study offers insight into the perceptions of cancer healthcare workers regarding psychedelic-assisted therapy; however, it is not without limitations. First, the study is limited in the ways that all qualitative approaches are limited. Although the interviewers endeavoured to present a neutral stance throughout, we note the influence of positionality and it is possible (if not likely) that unconscious cues supporting psychedelic research were conveyed. Additionally, as with most qualitative work, the sample was too small to fully capture the spectrum of possible perspectives or to quantify commonalities across participants [53]. Future work should assess whether the preliminary themes identified in the current work are confirmed in a larger sample and whether perspectives vary across healthcare workers ranging in age, profession, ethnicities and levels of experience. Nevertheless, there was consistency in views in the current work and, as with the theory of data saturation, the laws of diminishing returns were such that little new information surfaced in later interviews. Although it is beyond the scope of the current work to speculate how widely such perspectives might be held, future research is required to quantify the prevalence of such views and to determine whether these perspectives hold across geographical boundaries. Although we found no overt differences in the perceptions of participants across demographic or professional variables as identified in previous work [38], quantification of larger samples is required to determine such differences.

There are other methodological features of this study that require noting. As discussed earlier, the settings in which the interviews were conducted appeared to influence initial responses by some participants. Although standardising interview settings may help to minimise variation in responding, arguably, there was an important learning in noting this impact. As discussed earlier, a tension appeared between participants’ professional role and personal identity when talking about psychedelics. Although this tension was not directly articulated by participants, it was observed nonetheless. The boundaries of professionalism influenced our decision not to specifically ask participants about their personal experiences with psychedelics. Although some participants offered information about their personal experiences of their own accord, understanding how experience varied across participants may have further enriched our understanding of the data.

In considering future directions for research, it is also useful to consider other populations where an understanding of perspectives may inform the development of interventions. Important stakeholders include cancer patients themselves and their caregivers, palliative patients more generally, and consultation with indigenous practitioners, spiritual leaders, and cultural groups. This study highlighted the importance that future work be appropriately respectful of cultural aspects and indigenous knowledge. There is a long history of psychedelic use and psychedelic states of consciousness, whether induced by exogenous compounds or by practices such as fasting, chanting or dancing in many indigenous traditions. It is therefore critical that such traditions are given careful consideration and consultation is undertaken as a key component of any work in the emerging field of psychedelic medicine. These various stakeholder groups deserve the focus of future research, which would no doubt identify where there are gaps in care and give insight into whether patients and others perceive a place for psychedelics in modern healthcare. Furthermore, the current work did not explicitly discuss the role of ‘spirituality’ or ‘mystical experiences’ which have been shown to be important components of the psychedelic-assisted therapy experience [9,29]. How spirituality might fit (or not) alongside current medical paradigms is a key consideration in the development of future clinical trials and raises the important question of how findings might translate to contemporary medical contexts.

## 5. Conclusions

Understanding the perspectives of cancer healthcare workers is an important first step in developing interventions that are responsive to the people who will ultimately need to translate research findings to clinical application. In the current work, cancer healthcare workers acknowledged the importance of alleviating suffering in advanced cancer patients and noted a professional responsibility in preventing harm in such patients. In general, healthcare workers were open to the concept of psychedelic-assisted therapy approach and viewed it as an innovative approach; however, this view was also met with caution and highlighted the need for further research to ensure efficacy and safety.

## Figures and Tables

**Figure 1 ijerph-18-08160-f001:**
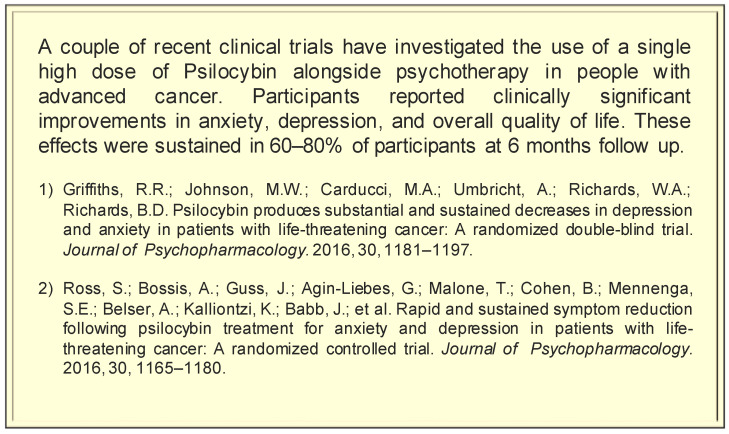
Card with summary of research investigating psychedelic-assisted therapy with cancer patients.

**Figure 2 ijerph-18-08160-f002:**
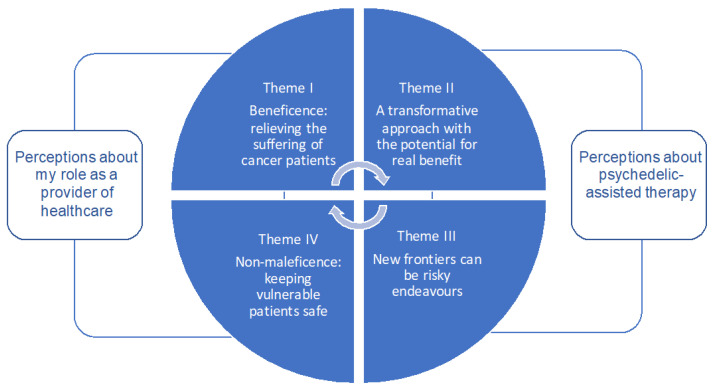
Key themes related to cancer healthcare workers perceptions of psychedelic-assisted therapy.

**Table 1 ijerph-18-08160-t001:** Participants’ demographics.

Participant Number	Age Range	Gender	Ethnicity	Profession	Years of Experience
1	40–49	Male	Asian	Doctor	>15 years
2	30–39	Female	NZ European	Doctor	5 to 15 years
3	30–39	Female	NZ European	Social worker	<5 years
4	20–29	Female	NZ European	Psychologist	<5 years
5	40–49	Female	NZ European	Social worker	>15 years
6	20–29	Female	Māori	Psychologist	<5 years
7	40–49	Female	NZ European	Psychologist	>15 years
8	50–59	Female	Māori	Psychologist	5 to 15 years
9	20–29	Female	Asian	Doctor	<5 years
10	30–39	Female	NZ European	Nurse	5 to 15 years
11	60–69	Male	NZ European	Doctor	>15 years
12	50–59	Male	Māori	Doctor	5 to 15 years

## Data Availability

For reasons of confidentiality, data will be available to investigators who provide a study plan and sign a confidentiality agreement with the University of Auckland. Requests should be submitted to L.M.R. and will not be unreasonably denied.

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
