# Peer review of "Cancer Healthcare Workers’ Perceptions toward Psychedelic-Assisted Therapy: A Preliminary Investigation"

_ijerph, 2021, doi:10.3390/ijerph18158160_

Round 1

Reviewer 1 Report

The current study aimed to understand the perceptions of cancer healthcare professionals about the potential use of psychedelic-assisted therapy in advanced cancer patients for benefiting the mental and physical pain relieve. The results demonstrated that cancer healthcare workers were open to the concept of psychedelic-assisted therapy approach and considered it as an innovative approach. They also appealed to further research on this therapy to ensure safety. The professional healthcare workers selected in this research strength more reliability of the conclusion.

Suggestions:

  1. Typo. Line 52 “debateable”
  2. Figure 2 is a little bit blurry.
  3. As is discussed in this article, whether more samples are needed to guarantee representativeness?
  4. Is it necessary to list each interviewees’ definite age?

Author Response

Thank you for the considered review of our manuscript. Please see our response to each of your points below.

Typo. Line 52 “debateable”

Thank you for alerting us to this error. This has now been corrected.

Figure 2 is a little bit blurry.

A clearer resolution figure has been added.

As is discussed in this article, whether more samples are needed to guarantee representativeness?

We have now elaborated on the need for larger samples to ensure representativeness as follows (lines 776-780): “Additionally, as with most qualitative work, the sample was too small to fully capture the spectrum of possible perspectives. Future work should assess whether the preliminary themes identified in the current work are confirmed in a larger sample and whether perspectives vary across healthcare workers ranging in age, profession, ethnicities and levels of experience.”

Is it necessary to list each interviewees’ definite age?

We have made the decision not to publish specific ages of participants for the purposes of preserving anonymity. The pool of cancer healthcare workers in New Zealand is not large and we believe that publishing age would unnecessarily risk identification of participants.

Reviewer 2 Report

This is a nicely written manuscript about the perceptions of cancer healthcare workers toward psychedelic-assisted therapy. However, there are concerns noted below for the authors' consideration in order to improve the quality of the paper.

Title: Consider deleting the phrase "What the buzz?" This appears to distract the reader from the topic of discussion, and has no bearing to the study.

Abstract: Line 14, revise the term "improving existential despair" to "ameliorating existential despair". 

Introduction: Line 52, delete "also", since you already stated "likewise". Line 87, give some examples of the dangers reported by the media.

Methods: Lines 118-122, move the narrative and table1 to the result section. Lines 138-140, the statements are confusing regarding the interviews. I believe that there were 12 study participants, and each of them was interviewed separately. Therefore, revise the statements to " Three participants were interviewed at their workplace, six at the university, two over the phone, and on at home." As is, the impression is that the 12 participants were all interviewed at their workplaces, the university, over the phone and at home. 

Results: Line 232, delete the phrase "As might be expected", and correct the English to "Participants described that their role as healthcare workers..."

Figure 2, consider labelling the themes within the chart : Themes I - IV.

Discussion and conclusions are satisfactory.

General comment: The term "maleficence" should be "non-maleficence". This should be corrected throughout the paper. Also, the authors did a very bad job of the citations. For example, in line 61, the citation [19] is missing, and in line 78, the citation jumps from [28] to [31], and then back to [30-32]. The citations throughout the paper should be revised.

Best of luck!

Author Response

Thank you very much for reviewing our work. We have responded to each of the issues you have raised and discuss these further below.

Title: Consider deleting the phrase "What the buzz?" This appears to distract the reader from the topic of discussion, and has no bearing to the study.

We have changed the article title with this comment in mind and also to address other concerns that we needed to more clearly identify that this study is preliminary work in the area. The title is now “Cancer healthcare workers’ perceptions toward psychedelic-assisted therapy: A preliminary investigation”

 Abstract: Line 14, revise the term "improving existential despair" to "ameliorating existential despair". 

Thank you for this suggestion. This term has been revised as suggested.

Introduction: Line 52, delete "also", since you already stated "likewise".

Deleted.

Line 87, give some examples of the dangers reported by the media.

We have revised this sentence to say “there have also been widespread reports in the media about dangers of psychedelic-induced violence and psychosis”

Methods: Lines 118-122, move the narrative and table1 to the result section.

Moved to the Results section as suggested.

Lines 138-140, the statements are confusing regarding the interviews. I believe that there were 12 study participants, and each of them was interviewed separately. Therefore, revise the statements to " Three participants were interviewed at their workplace, six at the university, two over the phone, and on at home." As is, the impression is that the 12 participants were all interviewed at their workplaces, the university, over the phone and at home. 

We have revised this statement as suggested.

Results: Line 232, delete the phrase "As might be expected", and correct the English to "Participants described that their role as healthcare workers..."

Thank you. This has been revised.

Figure 2, consider labelling the themes within the chart : Themes I - IV.

We have relabelled the themes from I-IV on Figure 2 as suggested.

Discussion and conclusions are satisfactory.

Thank you.

General comment: The term "maleficence" should be "non-maleficence". This should be corrected throughout the paper.

A very good point! This has been changed throughout.

Also, the authors did a very bad job of the citations. For example, in line 61, the citation [19] is missing, and in line 78, the citation jumps from [28] to [31], and then back to [30-32]. The citations throughout the paper should be revised.

Apologies for the problems with the citations. These have now been corrected.

Reviewer 3 Report

Very interesting read. I do not feel qualified to make comments or suggestions however, I would be interested to see if this improves our cancer patient's quality of life. It is always so depressing knowing you cannot save your patient. It boggles my mind how my patients, at the end of their life, with maybe 3 months to live.. live those last three months (including ones without much pain) extremely depressed. Each day is a blessing... given a death sentence or not... each day should be enjoyed.. especially knowing it could be your last.

Author Response

Thank you for your comments. Our intention for this work is to prompt discussion and thinking in this area and we appreciate you taking the time to note these.

Reviewer 4 Report

Comments for Manuscript Number ijerph-1302923entitled “What’s the buzz? A qualitative investigation of cancer healthcare workers’ perceptions toward psychedelic-assisted therapy" by L.M. Reynolds and co-authors.

 GENERAL COMMENTS

The paper of Reynolds et al. reports on a qualitative investigation through the interview with 12 healthcare professionals working in cancer care or palliative care. The study shows the perception of the patients psychological difficulties dealing with cancer. The psychedelic assisted therapy has been discussed. I do not recommend the publication of this paper in the International Journal of Environmental Research and Public Health (IJERPH); as major flaws in the conduction of the analysis have been described according to the comments that follow:

  • The authors should better describe in the introduction the main results of the pharmacological effects as well as side effects of psychedelic therapy.
  • The focus of the research is related to the healthcare professionals ‘perception relating their interviews: the research described the participants own replies which should rather be analyzed according to well established objectives.
  • The analysis is only conducted on 12 healthcare professionals, the classification according to the position, age, gender is not clearly established.
  • The data analysis section should be more centered on the methodological approach and the references related to the conduction of such analysis.
  • The described data analysis could be listed at the end of the manuscript describing the role of the authors.
  • The analysis is not conclusive and should be extended to a higher number of participants and a survey could be conducted on different levels instead of only reporting the participant’s comments.

Best Regards

Author Response

Thank you for your considered review of our manuscript. We have detailed below our response to each of your points and believe the manuscript is much improved as a result. 

The authors should better describe in the introduction the main results of the pharmacological effects as well as side effects of psychedelic therapy.

We have expanded on our discussion of the pharmacological effects of psychedelics as follows (lines 61-65): “The theoretical basis of this approach lies in the fact that classical serotonergic psychedelics, including psilocybin and lysergic acid diethylamide (LSD), appear to stimulate 5-HT2A receptors in the brain [18] and have been shown to increase synaptic function [19] and proteins that are associated with neural plasticity [Brain Derived Neurotrophic Factor, BDNF; 20]. Evidence suggests that 5-HT2A receptors are centrally involved in affect regulation [18].

We have also added more detail about the side effects of psychedelic therapy in the following statement (lines 85-87) “research investigating psychedelic agents has reported only mild or transient side effects including increased blood pressure and heart rate, anxiety, headaches and nausea [18].”

The focus of the research is related to the healthcare professionals ‘perception relating their interviews: the research described the participants own replies which should rather be analyzed according to well established objectives.

Given the preliminary and exploratory nature of this research our objectives were necessarily broad. We have added further detail about our objectives in the introduction which we hope provide greater clarity (lines 99-114) – “The objectives of the current work were to investigate the awareness, knowledge, attitudes, and perceptions of cancer healthcare workers in relation to the potential use of psychedelic-assisted therapy with advanced cancer patients who have depression and/or anxiety.

The analysis is only conducted on 12 healthcare professionals, the classification according to the position, age, gender is not clearly established.

In this preliminary work, our small sample precludes our ability to draw conclusions about how position, age and gender might influence perspectives. However, we agree that understanding how clinical and demographic factors might influence perspectives would be worthy of further consideration. We have now added a sentence in the section on study limitations that states that “future work should assess whether the preliminary themes identified in the current work are confirmed in a larger sample and whether perspectives vary across healthcare workers ranging in age, profession, ethnicities and levels of experience.” (lines 557-560).

The data analysis section should be more centered on the methodological approach and the references related to the conduction of such analysis.

We have expanded our methodological approach in the section on data analysis (lines 185-189) so that it more fully explains our approach and cites the appropriate literature on critical realist perspective (J. Maxwell, “A critical realist perspective for qualitative research,” in Qualitative Enquiry - Past, Present and Future: A Critical Reader, N. . Denzin and M. . Giardina, Eds. Walnut Creek, CA: Left Coast Press, Inc, 2015) and thematic analysis (V. Braun and V. Clarke, “Using thematic analysis in psychology,” Qual. Res. Psychol.).

The described data analysis could be listed at the end of the manuscript describing the role of the authors.

We appreciate that there was repetition in the description of data analysis and author contributions. We have now removed the author roles from the section on data analysis and included this information in the author contributions at the end of the manuscript.

The analysis is not conclusive and should be extended to a higher number of participants and a survey could be conducted on different levels instead of only reporting the participant’s comments.

We agree with this comment and have made several revisions to our manuscript to address this concern. These include:

  • Changing the title to emphasise the preliminary nature of this work: “Cancer healthcare workers’ perceptions toward psychedelic-assisted therapy: A preliminary investigation”.
  • Strengthening our comments about the limitations of the work (as noted above) so that we now state (lines 556-560) “the sample was too small to fully capture the spectrum of possible perspectives. Future work should assess whether the preliminary themes identified in the current work are confirmed in a larger sample and whether perspectives vary across healthcare workers ranging in age, profession, ethnicities and levels of experience.” We also note that “future work is required to quantify the prevalence of such views.” (lines 563-564)

Round 2

Reviewer 2 Report

The authors have responded adequately to my concerns from the initial review.

Author Response

Thank you again for your further review of our work. Your careful consideration of the manuscript is very much appreciated. 

Reviewer 4 Report

GENERAL COMMENTS

The paper of Reynolds et al. reports on a preliminary investigation through the interview with 12 healthcare professionals working in cancer/palliative care on their perception of psychedelic assisted therapy. The review has been improved, and the authors took into consideration some recommendations. The change of the title is a good initiative. I recommend the publication of this paper in the International Journal of Environmental Research and Public Health (IJERPH); with major revision specially in the report of the participants replies according to the comments that follow:

  • The research could bring a different way to analyze the participants replies based on the used vocabulary, main concerns, and within charts or an assessment of the common features between all the answers.
  • The authors explained the reason why the analysis was only conducted on 12 healthcare professionals, but they could bring these results into the establishment of a new survey and a different evaluation that could be largely diffused to a higher number and provide statistically more rugged analysis (maybe beyond the geographical localization).

Author Response

We are very grateful for this reviewer's further consideration of our work and suggestions for improvement. We have addressed each of the reviewer's comments below with the reviewer's comments in bold and our responses in italics:

The research could bring a different way to analyse the participants replies based on the used vocabulary, main concerns, and within charts or an assessment of the common features between all the answers.

We agree that there may be different ways to assess the common features between participants and to approach the analysis of qualitative data. Our approach has been based on recommendations by highly-regarded qualitative experts who argue that quantifying qualitative data from small samples is problematic and runs the risk of credentialing counting (for more information on this see: Monrouxe, L. V., & Rees, C. E. (2020). When I say … quantification in qualitative research. Medical Education, 54(3), 186–187; Hannah, D. R., & Lautsch, B. A. (2010). Counting in Qualitative Research: Why to Conduct it, When to Avoid it, and When to Closet it. Journal of Management Inquiry, 20(1), 14–22.). However, we agree that some readers might raise this question and so have now expanded our comments on this possibility under the section on study limitations stating that “the sample was too small to fully capture the spectrum of possible perspectives or to quantify commonalities across participants [53]” (p.11).

The authors explained the reason why the analysis was only conducted on 12 healthcare professionals, but they could bring these results into the establishment of a new survey and a different evaluation that could be largely diffused to a higher number and provide statistically more rugged analysis (maybe beyond the geographical localization).

We agree that the obvious next step in the progression of this research is to conduct a larger survey that extends beyond the geographical boundaries of the current work and quantifies the prevalence of the themes using statistical analytical approaches. We are in the process of conducting such a study and intend submitting that for publication on completion of our data collection and manuscript write-up. We have now added a point for clarity in the Discussion that states “future work is required to quantify the prevalence of such views and to determine whether these perspectives hold across geographical boundaries” (p.11).